# Genome-Wide Identification and Characterization of *USP* Gene Family in Grapes (*Vitis vinifera* L.)

Tao Xu [1,†] , Tianchi Chen [1,†] , Tianye Zhang [2] , Leyi Shen [1], Zhe Chen [1], Yue Xu [1], Yueyan Wu [1,*] and Jian Yang [2,*]

[1]  College of Biological and Environmental Sciences, Zhejiang Wanli University, Ningbo 315100, China
[2]  State Key Laboratory for Quality and Safety of Agro-Products, Institute of Plant Virology, Ningbo University, Ningbo 315211, China
*  Correspondence: wyynb2009@163.com (Y.W.); nather2008@163.com (J.Y.); Tel.: +86-0574-8822-2235 (Y.W.)
†  These authors contributed equally to this work.

**Abstract:** With the frequent occurrence of extreme natural disasters, unfavorable growth environment is a common phenomenon in the life cycle of plants. In recent years, universal stress proteins (USPs) have attracted extensive attention in the field of plant science for their expression patterns and functional analysis. However, the characterization of the USP family remains unclear in grape. In this study, we identified 30 *VvUSPs* in the grape genome, which could be divided into three classes according to their encoded protein sequences, and this classification was reflected by the distribution of conserved motifs. Gene duplication analysis demonstrated that segmental duplication was an important pathway in the expansion of the *VvUSP* family. The expression patterns of 12 *VvUSPs* were significantly different between tissues, implying they had different functions in various tissues. *Cis*-acting element and expression analysis showed that most of the promoter regions of *VvUSPs* contained sequences responsive to hormones and stress elements, especially the promoter region of VIT_16s0013g01920. In conclusion, our findings provide comprehensive information for the further investigation of the genetics and protein functions of the *USP* gene family in grape.

**Keywords:** grape; universal stress protein; abiotic stress; hormone

## 1. Introduction

In recent years, the intensity, frequency and duration of extreme weather events have been altered by either natural or anthropogenic impacts, with effects on environmental factors affecting plant growth, such as temperature [1], water [2], shade [3] and salt content [4]. In particular, global warming can lead to desertification, drought and the intensification of soil salinity, which adversely affect the quality and yield of crops [5,6]. In fact, it is impossible for plants to completely avoid unfavorable conditions for growth [7]. Therefore, it is important to explore the impacts of environmental stress on seed germination and plant development, as well the associated regulation of gene expression, and the stimulation of other physiological reactions [8].

When plants are subjected to abiotic stress, their deployment of stress resistance genes can mitigate the effects of environmental stress, but the identities and/or roles of some of these genes remain unclear [9,10]. Universal stress proteins (USPs) were first characterized in *Escherichia coli* and could be divided into the USP subfamilies A–G [11,12]. *USPA* genes and their proteins were later detected in many phylogenetically diverse organisms, including plants. The function of protein kinase containing USPAs in grape and Arabidopsis has been reported previously [13,14]. Although the regulation of plant *USPA* genes has been studied to some extent, the roles of *USP* gene family members in some species remain unclear and require further study.

Many proteins containing at least one USP domain have been found in many organisms, among which some contain α/β subdomain structure [11,15]. USP proteins act alone

or in combination with interacting partners [16]. In addition, there are also some plant USPs showing nucleic acid repair and refolding activities, which contribute to the reduction of nucleic acid structural damages under stress conditions. It has been previously proved that USP inhibition decreases transforming growth factor-β (TGF-β) signaling in primary fibroblast cell lines, including USP15-enhanced TGF-β-induced fibrosis response, which then protects plant growth under abiotic stress [17]. USPs represent the largest family of ubiquitinases. However, ubiquitination is an important process to regulate cell metabolism and the cell cycle, and the ubiquitination of common target proteins can lead to protein degradation by proteasomes [18,19]. Therefore, a better understanding of USPs may assist in identifying targeted breeding strategies to improve crop performance under suboptimal environmental conditions.

Phytohormones, such as gibberellin (GA), salicylic acid (SA), abscisic acid (ABA) and methyl jasmonate acid (MeJA) play important roles in regulating key aspects of plant growth, development and stress responses [20–22], including abiotic stresses from unfavorable environmental conditions [23,24]. Previous studies have shown that the functional activity of several Arabidopsis *USP* genes could be directly or indirectly related to phytohormone-dependent processes.

Grapevine (*Vitis vinifera* L.) is one of the economically important crops in the world, and its yield and quality are related to the living standards and economic development of farmers [25]. However, grapes are often under abiotic stress due to uncontrollable factors in the vineyard environment, which severely limits their yield and quality [26–28]. In the main grape-producing areas of southern China, the accumulation of grape nutrients and the grape quality in the summer are seriously affected by the high temperatures and the rain season. In addition, incidences of sea incursions on the Chinese middle east coastline has caused soil salinization, with deleterious effects on grape growth and yield [29]. In this study, we used a bioinformatics approach to identify members of the *VvUSP* gene family and to explore their conserved motifs, *cis*-regulatory elements, and those having undergone gene duplication events. In addition, the expression patterns of some family members of *VvUSP* were examined in different tissues, after exogenous hormone treatment and under conditions of abiotic stresses. This study describes the systematic characterization of *VvUSPs*, which will assist in future research aimed at assigning function to grapevine USPs.

## 2. Materials and Methods

### 2.1. Identification of USP Gene Family in Grape

To identify USP genes in the grape genome and study the relationship between VvUSPs and AtUSPs, we downloaded the genome and protein sequences of grape (GCA_000003745.2) and Arabidopsis (GCA_000001735.1) from the Ensembl plants database (http://plants.ensembl.org/; accessed on 17 June 2022). The hidden Markov model profile of the USP conserved domain (PF00582) was downloaded from the Pfam protein families database (http://pfam-legacy.xfam.org/; accessed on 17 June 2022) and used to screen the grape and Arabidopsis genomes to identify USP proteins. All results were further screened by the Pfam and SMART databases (http://smart.embl.de/ accessed on 30 June 2022) and NCBI-CDD (https://www.ncbi.nlm.nih.gov/cdd/; accessed on 30 June 2022). Nonredundant USP protein sequences were used in all further analyses. In addition, the length of sequences, molecular weights and theoretical isoelectric point of the deduced polypeptide of USP protein in grapes were predicted by the ExPasy website (http://web.expasy.org/protparam/ accessed on 1 June 2022). SoftBerry (http://www.softberry.com/ accessed on 7 July 2022) was used to predict protein subcellular localization.

### 2.2. Phylogenetic Analysis, Gene Structure and Conserved Motif Analysis

Multiple sequence alignments of VvUSP and AtUSP protein sequences and construction of phylogenetic trees were performed using MEGA6 and the neighbor-joining method (1000 bootstrap repeats) [30]. cDNA sequences of *VvUSPs* were aligned in GFF files for to obtain exon-intron, coding sequence (CDS) and untranslated region (UTR) data using GSDS

(http://gsds.gao-lab.org/ accessed on 23 July 2022) [31]. TBtools was used to identify conserved motifs of protein sequences in all VvUSPs [32].

### 2.3. Chromosomal Location, Gene Duplication and Synteny Analysis

The physical distribution and locations of the *VvUSP* genes on chromosomes were drafted with MapChart. The coding sequences of the duplicated genes were aligned using ClustalW, and the values of *Ka* (nonsynonymous substitution rate) and *Ks* (synonymous substitution rate) were calculated based on the alignment result using the *KaKs*_calculator script [33]. The approximate date of duplication events was estimated using the formula T = *Ks*/2R, where R represents the rate of divergence for the nuclear gene. The R was assumed to be $1.5 \times 10^{-8}$ synonymous substitutions per gene per year in dicotyledonous plants [34]. MCScanX was used to detect the commonality of *USP* genes between grape and other species.

### 2.4. Cis-Acting Regulatory Element Analysis

The PlantCARE database (http://bioinformatics.psb.ugent.be/webtools/plantcare/html/ accessed on 28 July 2022) was used to identify *cis*-acting regulatory elements in the promoter regions (from 1500 bp upstream of the start codon) of each *VvUSPs* [35].

### 2.5. Plant Materials and Treatments

To explore the gene expression patterns in different tissues, tendril, leaf, stem, flower, root, seed, grape flesh and grape skin were harvested from 2-month-old 'YinHong' grape plants grown in a greenhouse at Zhejiang Wanli University. All seedlings were grown in a controlled artificial climatic chamber under the same conditions (16 h light at 25 °C/8 h dark at 18 °C, 60% relative humidity). Three biological replicates were sampled after plants reached the age of 2 months. The hormone treatment involved spraying the plants with 200 μM SA, 200 μM MeJA (Aladdin Biochemical Technology, Shanghai, China), or distilled water as the control. Triplicate leaf samples were then collected randomly at five time intervals (5 min and 2, 4, 6 and 8 h). The shade-stress treatment used 0, 1 (30%), 2 (60%) and 3 (90%) layers of sunshade net to restrict the sunlight on the plants. Leaf samples were collected randomly in triplicate 20 and 40 days after treatment. The salt-stress treatment was established by drip-irrigating the plants with 0, 2 (0.2%), 4 (0.4%) and 6 g·L$^{-1}$ (0.6%) sodium chloride solution, and randomly selected leaf samples were collected 20 and 40 days after treatment. For the heat-stress treatment, the plants were exposed to 35 °C and leaves were sampled at 0, 3 and 6 h after treatments. All the samples were frozen by liquid nitrogen and stored at −80 °C until total RNA was extracted.

### 2.6. RNA Extraction and qRT-PCR Analysis

A HipPure Plant RNA Mini Kit (Magen, Guangzhou, China) was used to extract the total RNA of each leaf sample as per the manufacturer's instructions. The first-strand cDNA Synthesis SuperMix (Novoprotein, Suzhou, China) was used to synthesize the first-strand cDNA. The expression levels of 12 *VvUSPs* were determined using SYBR qPCR SuperMix Plus (Novoprotein). The 10 μL qPCR reaction system was: 5 μL Mix enzyme, 0.2 μL upstream primer, 0.2 μL downstream primer, 0.2 μL cDNA, 4.4 μL $H_2O$. The qPCR reaction program was: 95 °C for 1 min; 95 °C for 20 s, 60 °C for 1 min, 45 cycles of these conditions; 65 °C for 5 s; 0.5 °C·s$^{-1}$ to 95 °C for dissolution curve analysis. The primers used in qRT-PCR analyses were designed using Primer 5 software, and are listed in Table S1. The $2^{-\Delta\Delta Ct}$ method was used to calculate the relative expression level of the 12 *VvUSPs*. The significance of differences was calculated using SPSS statistics 25 (IBM, Armonk, NY, USA), and the histogram was drawn using Origin2021 (OriginLab, Northampton, MA, USA).

## 3. Results

### 3.1. Identification and Analysis of VvUSPs

Thirty potential USP sequences were identified in the grape genome and their gene ID, CDS length, protein size, theoretical pI and molecular weight subcellular localization are shown in Table 1 and Table S2. The lengths of VvUSP proteins ranged from 129 (VIT_08s0032g00590) to 797 (VIT_04s0008g02910) amino acids with molecular weights of 14.2 kDa (VIT_08s0032g00590) to 89.1 kDa (VIT_04s0008g02910). The pI values of VvUSPs ranged from 4.71 (VIT_00s0203g00200) to 10.93 (VIT_17s0000g04980) with an average of 7.78. Among the 30 VvUSP proteins, 10 were predicted to be cytoplasmic and 11 to be mitochondrial; 2 were located in the nucleus and 2 were in the plasma membrane.

**Table 1.** Gene ID, CDS length, protein size, and the prediction of theoretical pI, molecular weight and subcellular localization for each *VvUSP*.

| Gene Name | CDS Length | Protein Size | Theoretical pI | Molecular Weight | Subcellular Localization |
|---|---|---|---|---|---|
| VIT_01s0011g00080 | 678 | 225 | 10.39 | 25,021.2 | Mitochondrial |
| VIT_01s0011g03680 | 513 | 170 | 8.6 | 18,218.9 | Extracellular |
| VIT_01s0011g03690 | 495 | 164 | 7.01 | 18,070.7 | Plasma membrane |
| VIT_01s0011g03700 | 1092 | 363 | 8.16 | 40,109.7 | Plasma membrane |
| VIT_03s0038g04750 | 522 | 173 | 5.64 | 18,899.7 | Cytoplasmic |
| VIT_04s0008g01360 | 2199 | 732 | 8.67 | 81,742.4 | Golgi |
| VIT_04s0008g02910 | 2394 | 797 | 7.27 | 89,051.2 | Cytoplasmic |
| VIT_04s0079g00610 | 495 | 164 | 7.29 | 17,816.4 | Mitochondrial |
| VIT_05s0020g01190 | 477 | 158 | 8.02 | 17,088.6 | Cytoplasmic |
| VIT_05s0049g02120 | 600 | 199 | 10.43 | 21,761 | Mitochondrial |
| VIT_05s0077g01070 | 498 | 165 | 5.82 | 18,055.7 | Cytoplasmic |
| VIT_06s0004g05730 | 495 | 164 | 6.79 | 18,014.6 | Mitochondrial |
| VIT_07s0005g00290 | 1014 | 337 | 7.9 | 36,033.5 | Cytoplasmic |
| VIT_07s0005g01350 | 432 | 143 | 8.3 | 15,577 | Cytoplasmic |
| VIT_08s0007g01430 | 495 | 164 | 7.51 | 18,238.8 | Mitochondrial |
| VIT_08s0032g00590 | 390 | 129 | 8.65 | 14,197.5 | Extracellular |
| VIT_09s0054g01060 | 2340 | 779 | 7.26 | 87,164.3 | Mitochondrial |
| VIT_10s0116g00430 | 828 | 275 | 5.11 | 30,278.2 | Cytoplasmic |
| VIT_11s0016g01830 | 2310 | 769 | 7.06 | 86,112.4 | Mitochondrial |
| VIT_11s0065g00390 | 2226 | 741 | 8.1 | 82,191 | Golgi |
| VIT_12s0134g00430 | 762 | 253 | 5.18 | 27,630.7 | Nuclear |
| VIT_14s0060g01300 | 513 | 170 | 7.41 | 18,301.1 | Mitochondrial |
| VIT_14s0060g01320 | 612 | 203 | 7.25 | 22,214.7 | Cytoplasmic |
| VIT_14s0066g01130 | 528 | 175 | 6.62 | 19,891.7 | Cytoplasmic |
| VIT_16s0013g01920 | 2355 | 784 | 7 | 87,702.1 | Cytoplasmic |
| VIT_17s0000g04260 | 633 | 210 | 10.19 | 23,398.6 | Nuclear |
| VIT_17s0000g04980 | 696 | 231 | 10.93 | 25,818.9 | Mitochondrial |
| VIT_18s0001g03000 | 660 | 219 | 9.94 | 23,863.5 | Mitochondrial |
| VIT_18s0001g07360 | 507 | 168 | 7.11 | 17,991.6 | Mitochondrial |
| VIT_00s0203g00200 | 465 | 154 | 4.71 | 16,917 | Nuclear |

### 3.2. Gene Structure and Conserved Motifs

In order to assess the evolutionary relationship among USPs from grape (30) and Arabidopsis (40), a phylogenetic tree was constructed (Figure 1). The tree shows that USP members of both spp. clustered into groups A, B, C and D.

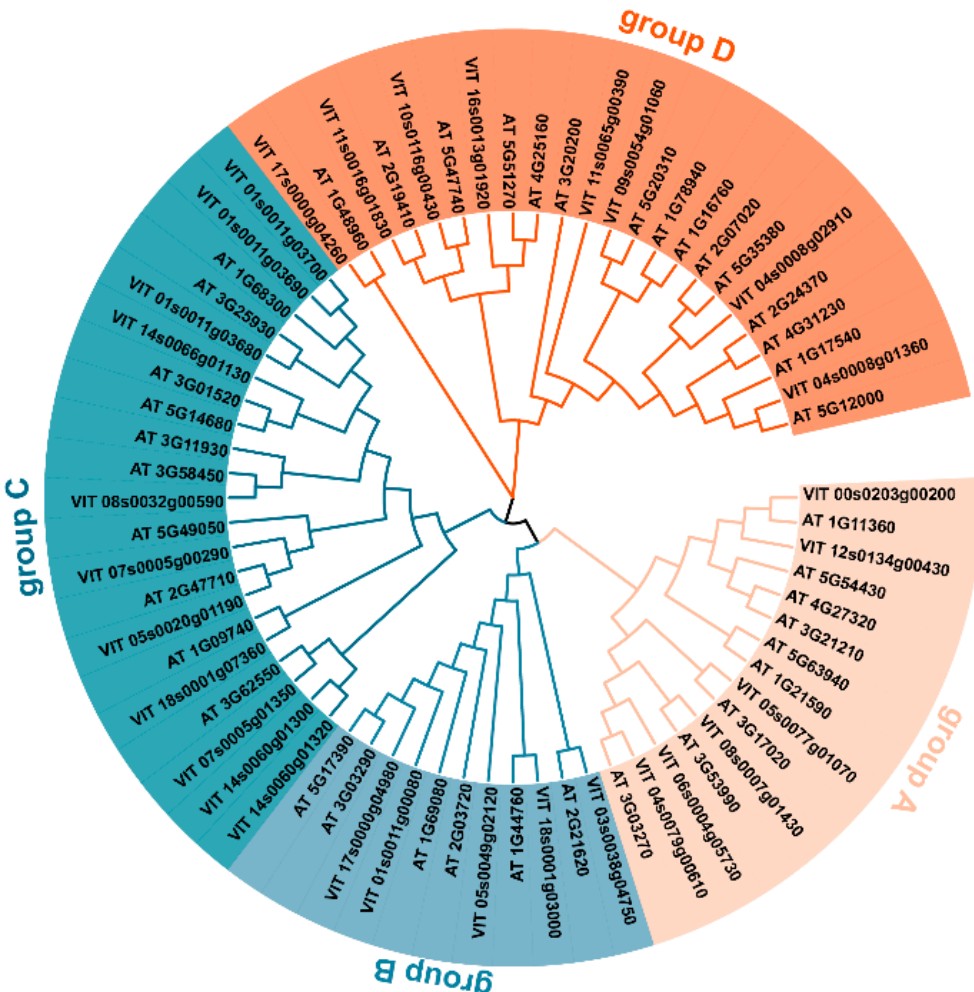

**Figure 1.** Phylogenetic analysis of VvUSPs and AtUSPs.

In order to assess whether the hierarchical clustering observed (Figure 1) reflected differences in USP functionalities, a phylogenetic tree of 30 VvUSPs (Figure 2a) was constructed and superimposed on conserved motif content (b) and gene structure (c). Ten conserved motifs were found to be present among the 30 VvUSP proteins (Figure S1 and Table S3), but their distribution varied between the different VvUSPs, which contained from only one to up to seven conserved motifs (Figure 2b). With the exception of VIT_07s0005g01350 and VIT_08s0032g00590 of subclass Ib, all VvUSP proteins contained motif 4. However, motifs 1, 7 and 9 were restricted to classes I and III, whereas motifs 2, 3, 5, 6, 8 and 10 were present only in class II. In general, the distribution of conserved motifs can be seen to reflect the hierarchical clustering based on VvUSP primary sequences. The analysis of the gene structure (Figure 2c) shows that 29/30 *VvUSPs* contained identifiable intron regions. The exception to this was VIT_11s0065g00390, which may be caused by functional differentiation during evolution. In class I, most genes contain two intron regions and three to seven exons. In addition, the number of exons in class II *VvUSPs* is relatively high compared with the other classes.

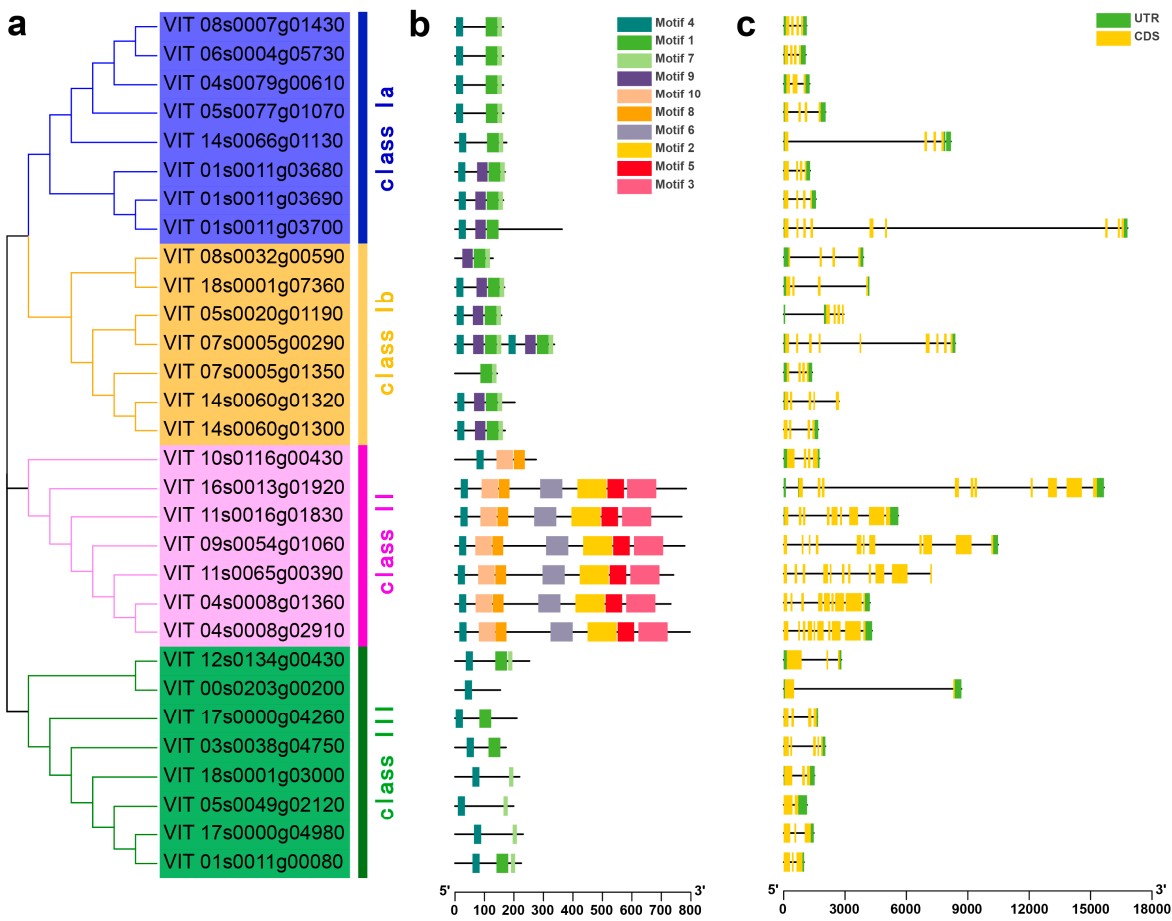

**Figure 2.** Phylogenetic analysis (**a**), motif analysis (**b**) and gene structure (**c**) of VvUSP protein family.

*3.3. Chromosomal Location and Collinearity Analysis*

As shown in Figure 3a, the 29 *VvUSP* genes are assigned to chromosomes 1, 3–12, 14, 16-18, and one unknown (VIT_00s0203g00200). The *USPs* showed an uneven distribution between chromosomes. A total of 4 *VvUSP* genes clustered within a short distance at the top of chromosome 1, whereas chromosomes 3, 6, 9, 12 and 16 each harbored a single *VvUSP* gene, and no *VvUSP* genes were present in chromosome 19. In addition, we found this distribution to be independent of chromosome length.

Based on a collinearity analysis of the *VvUSP* gene family using MCScanX, a total of six segmental-duplicated gene pairs were observed between Chr1 and Chr17 (VIT_01s0011g00080 and VIT_17s0000g04980), Chr4 and Chr11 (VIT_04s0008g01360 and VIT_11s0016g01830), Chr5 and Chr7 (VIT_05s0020g01190 and VIT_07s0005g00290), Chr6 and Chr8 (VIT_06s0004g05730 and VIT_08s0007g01430), Chr7 and Chr14 (VIT_07s0005g01350 and VIT_14s0060g01300) (Figure 3b). We found that part of the *VvUSP* gene comes from segmental replication, which indicates that fragment replication plays a major role in the expansion of the *VvUSP* gene family. The distributions of synonymous substitutions per site (*Ks* values) can be used to estimate the evolution time of whole genome duplications or segment repeat events [36]. Our results show that the *Ks* values of all gene pairs are between 0.69 and 4.43 (Table 2). Relatively speaking, the *Ks* value for *VvUSPs* is small, indicating a short evolution time for *VvUSPs*. We calculated that the approximate date of whole-genome or segmental repeats occurred from 22.97 Mya (*Ks* = 1.37) to 147.80 Mya (*Ks* = 0.03), with an average of 59.59 Mya (*Ks* = 1.79).

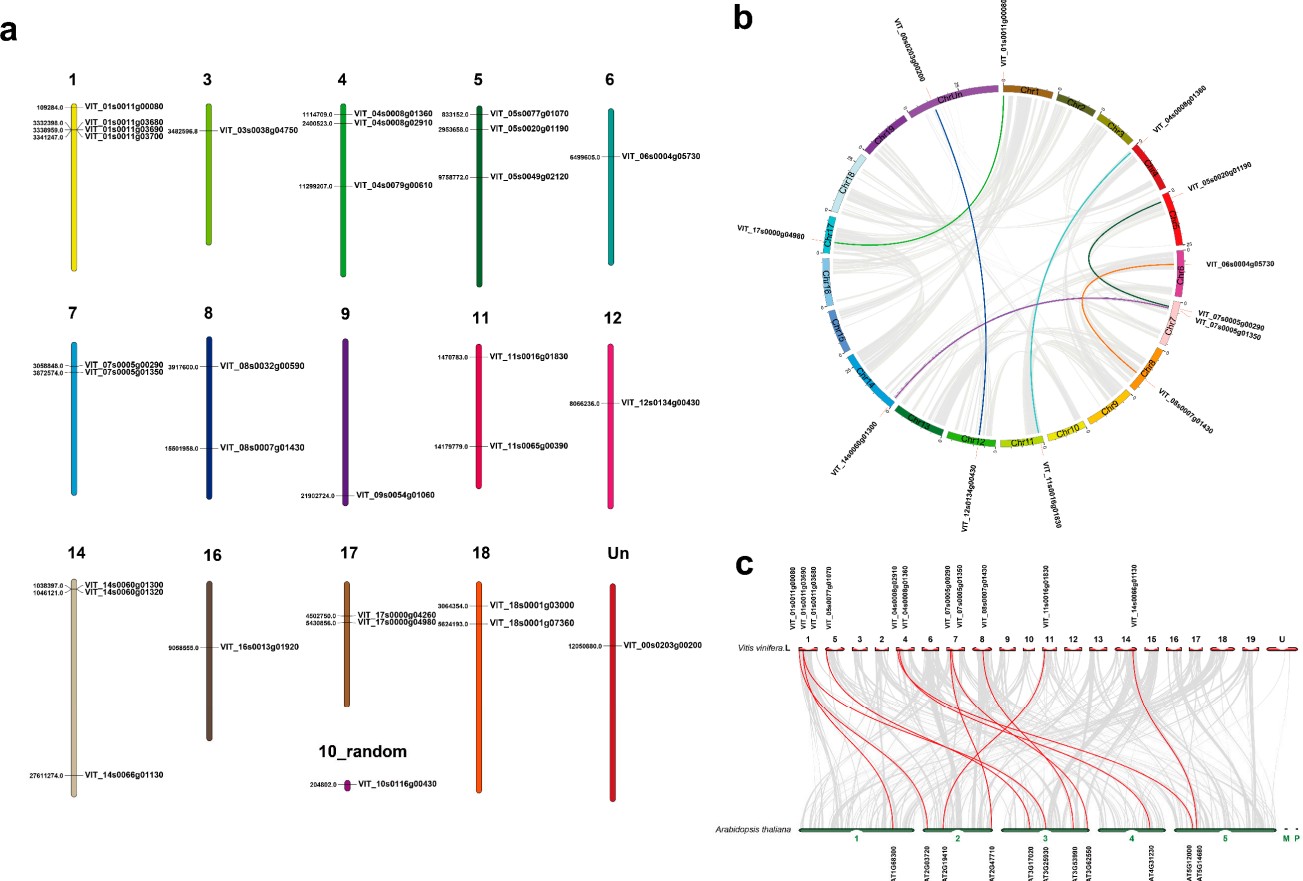

**Figure 3.** The chromosome distribution of *VvUSP* genes (**a**), collinearity analysis of *VvUSPs* (**b**) and collinearity analysis of *VvUSPs* and *AtUSPs* (**c**). The *USP* genes and *USP* homologs are represented in red and grey, respectively.

**Table 2.** Calculation of *Ka*, *Ks*, and *Ka/Ks* and divergent time of *VvUSP* gene pairs.

| Duplicated Gene Pairs | *Ka* | *Ks* | *Ka/Ks* | Duplicated Type | Divergence Time (Mya) |
|---|---|---|---|---|---|
| VIT_17s0000g04980/VIT_01s0011g00080 | 0.33 | 1.30 | 0.25 | Segmental | 43.44 |
| VIT_08s0007g01430/VIT_06s0004g05730 | 0.25 | 1.93 | 0.13 | Segmental | 64.28 |
| VIT_14s0060g01300/VIT_07s0005g01350 | 0.98 | 1.06 | 0.92 | Segmental | 35.43 |
| VIT_04s0008g01360/VIT_11s0016g01830 | 0.94 | 0.69 | 1.37 | Segmental | 22.97 |
| VIT_12s0134g00430/VIT_00s0203g00200 | 0.12 | 4.43 | 0.03 | Segmental | 147.80 |
| VIT_17s0000g04980/VIT_01s0011g00080 | 0.33 | 1.30 | 0.25 | Segmental | 43.44 |

　　　　To further examine the evolutionary mechanisms underlying the diversification of the *USP* family in grapevine, a comparative syntenic map was constructed with *USPs* from grapevine and Arabidopsis (Figure 3c). A total of 11 *VvUSP* genes showed a syntenic relationship with those in Arabidopsis between VIT_01s0011g03690 and AT1G68300, VIT_01s0011g00080 and AT2G03720, VIT_01s0011g03680 and AT3G25930, VIT_04s0008g02910 and AT4G31230, VIT_04s0008g01360 and AT5G12000, VIT_05s0077g01070 and AT3G17020, VIT_07s0005g00290 and AT2G47710, VIT_07s0005g01350 and AT3G62550, VIT_08s0007g01430 and AT3G53990, VIT_11s0016g01830 and AT2G19410, and VIT_14s0066g01130 and AT5G14680. This indicated that the evolutionary relationship between *VvUSP* and *AtUSP* was relatively close, which provided a reference for us to study the grape *USP* gene.

### 3.4. Cis-Acting Elements Present in the Promoter Regions of VvUSP Family Genes

Predicted *cis*-acting elements in *VvUSP* promoters were searched for within the first 1500 bp from the start codon of each *VvUSP* (Figure 4 and Table S4). Most grape *USP* promoters contain several *cis*-acting elements, reported to be involved in stress and hormone induction, including sequence motifs such as abscisic-acid-responsive element (ABRE), anoxia-responsive element (ARE), low-temperature response (LTR), the MYB binding site involved in drought responses (MBS), and defense and stress responsiveness (TC-rich), among which the ABRE and ARE *cis*-acting elements were the most common. In addition, we also found that nine *VvUSP* genes contain TC-rich elements. These results suggest that *VvUSP* genes may play a role in stress tolerance.

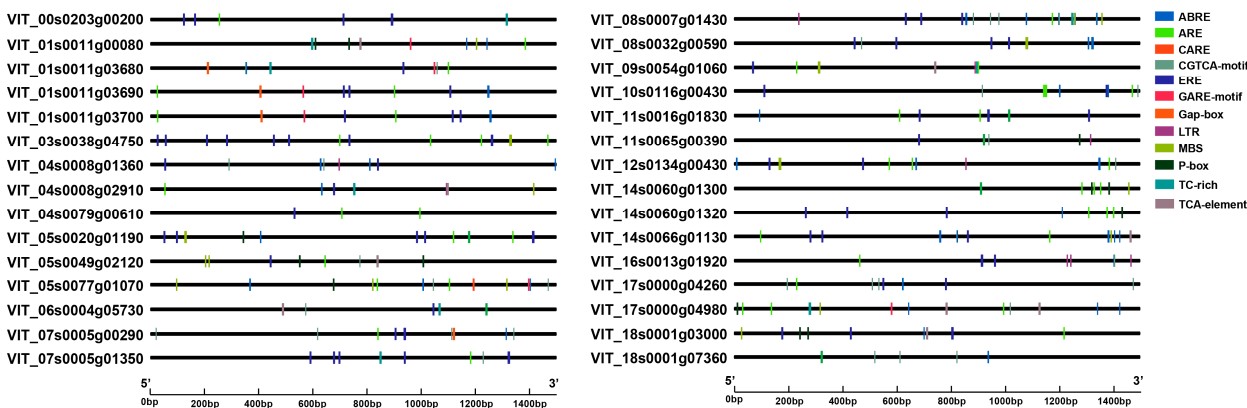

**Figure 4.** Promoter analysis of the *VvUSPs*. The promoter analysis of *VvUSPs* was performed on 1500 bp sequences upstream from their transcription start sites.

### 3.5. Tissue-Specific Analysis of VvUSPs

The expression patterns of 12 *VvUSP* genes randomly selected from each of class Ia, class Ib, class II and class III in eight tissues of grapevine and grape using qRT-PCR analysis are shown in Figure 5. We found that most *VvUSPs* showed a widely distributed expression between tissues, although the expression levels of 10/12 *VvUSPs* were relatively lower in skin. VIT_07s0005g01350 and VIT_12s0134g00430 were expressed significantly higher in seed, suggesting that the gene may be involved in seed germination or dormancy. Other examples of include VIT_01s0011g03680 and VIT_14s0066g01130 (stem), VIT_05s0077g01070, VIT_07s0005g00290 and VIT_18s0001g07360 (tendril), VIT_10s0116g00430, VIT_16s0013g01920 and VIT_17s0000g04260 (flower), and VIT_03s0038g04750 (leaf). These results indicate that some *VvUSPs* showed tissue-specific expressions. The expression patterns of *VvUSPs* within the same hierarchical clusters (Figure 2a) and with similar conserved domains (Figure 2b) showed significant differences in tissue expression, indicating these *VvUSPs* play different roles in different tissues during plant growth and development.

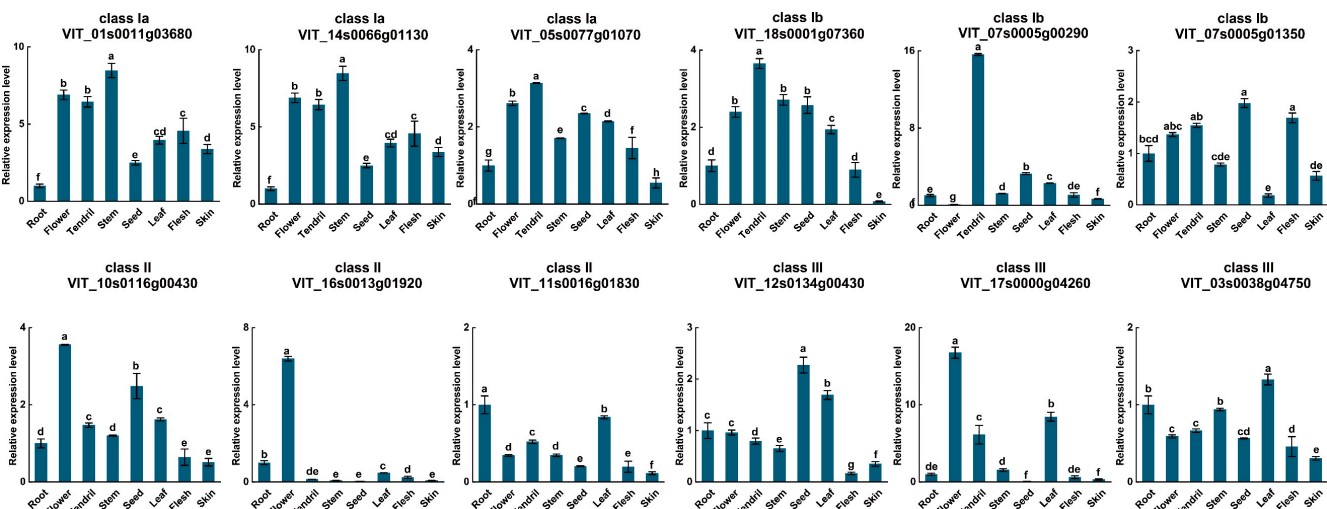

**Figure 5.** Tissue-specific analysis of 12 *VvUSP* genes by qRT-PCR. Different letters above the bars indicate a significant difference ($p < 0.05$).

### 3.6. Expression Profiles of VvUSPs in Response to Diverse Hormone Treatments

Plant hormones can regulate plant defense against various pathogens, such as SA and MeJA [37]. To explore the response of *VvUSP* family members to these hormones, we selected three *VvUSPs* from each of the classes Ia, Ib, II and III and their responses to hormone treatments were analyzed by qRT-PCR (Figure 6). With the exception of VIT_05s0077g01070, the expression of *VvUSPs* was seen to be sensitive to the hormone treatments (MeJA and SA), and showed significant alterations in relative expression levels over the course of the treatments. VIT_16s0013g01920 was up-regulated most obviously by SA 6–8 h after treatment, whereas similar changes in expression were observed for VIT_11s0016g01830 and VIT_12s0134g00430 at 8h. Significant increases in the expression of seven genes' *VvUSPs* (VIT_07s0005g00290, VIT_07s0005g01390, VIT_10s0116g00430, VIT_16s0013g01920, VIT_11s0116g01830, VIT_12s0134g00430 and VIT_17s0000g04260) could be detected as early as 5 min after SA treatment. However, these early increases were transient and preceded later increases, decreases or both. Under MeJA treatment, VIT_01s0011g03680 and VIT_16s0013g01920 showed a significant transient up-regulation (4–6 h and 2–4 h, respectively). The expression levels of VIT_01s0011g03680, VIT_07s0005g00290, VIT_07s0005g01350 and VIT_12s0134g00430 reached the maximum at 5 min, VIT_16s0013g01920 had the highest expression at 6 h. We observed that VIT_16s0013g01920 showed the highest sensitivity to MeJA treatment, which may be related to the LTR element in the promoter region of this gene.

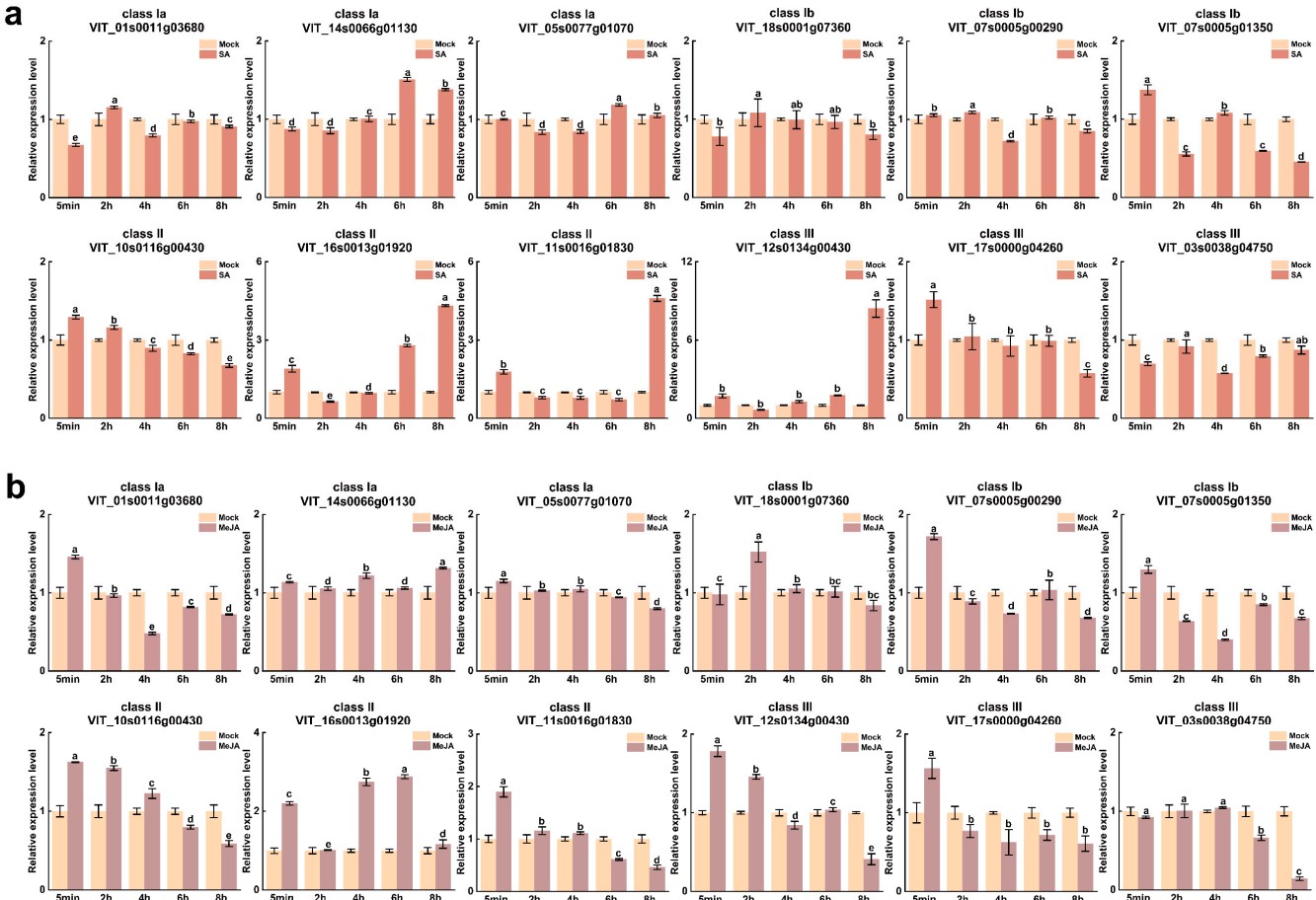

**Figure 6.** Expression levels of *VvUSP* genes under SA (**a**) and MeJA (**b**) treatments by qRT-PCR. Different letters above the bars indicate a significant difference ($p < 0.05$).

### 3.7. Analysis of VvUSP Expression under Abiotic Stress Treatments

The grape industry is seriously affected by abiotic stresses due to changes in the environment. In order to further study the effect of environmental stresses on *VvUSP* gene expression, we examined their expression patterns under shade, salt stress and heat stress. Twelve *VvUSP* genes were randomly selected from the four main phylogenetic groups and the effects of stress treatments on their expression levels were measured by qRT-PCR.

Light is an important environmental factor for plant photosynthesis and has an important impact on plant growth and berry quality. We observed that the expression levels of the *USP* genes under shade stresses at 20 and 40 days varied among the 12 members (Figure 7). Interestingly, two *VvUSP* genes (VIT_16s0013g01920 and VIT_11s0016g01830) were up-regulated after 20 and 40 days of the 30% and 60% shade-stress treatment. In addition, seven *VvUSP* genes (VIT_16s0001g07360, VIT_07s0005g01350, VIT_10s0116g00430, VIT_16s0013g01920, VIT_11s0016g01830, VIT_17s0000g04260 and VIT_03s0038g04750) all showed up-regulation after 20 days under 60% shade-stress treatment. VIT_17s0000g04620 was seen to be the most sensitive to the shade treatment, followed by VIT_16s0013g01920 and VIT_11s0016g01830, which suggested they might play an essential regulatory role in the response to low light stress.

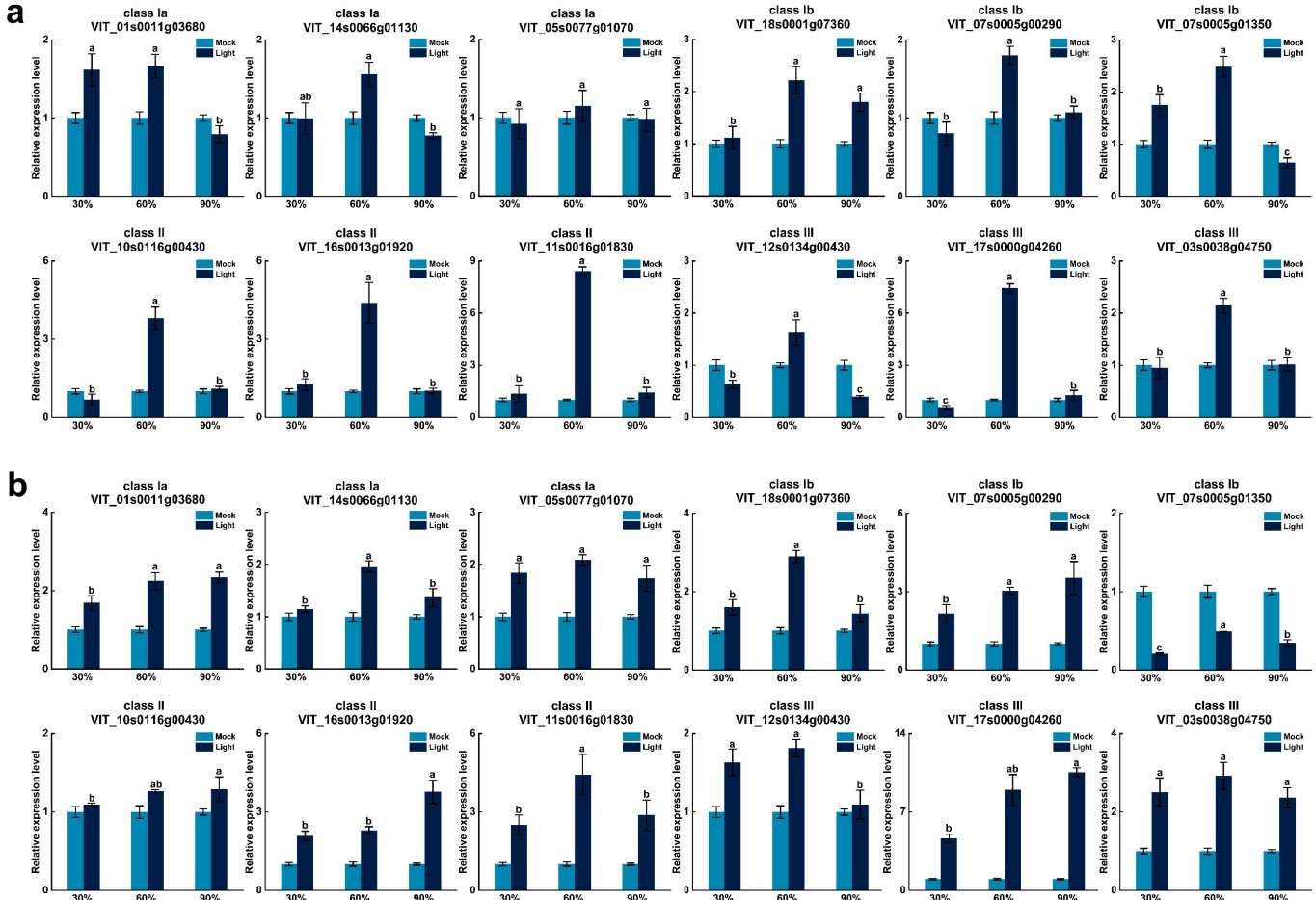

**Figure 7.** Expression levels of *VvUSP* genes at 20 days (**a**) and 40 days (**b**) of low-light-stress treatments by qRT-PCR. Different letters above the bars indicate a significant difference ($p < 0.05$).

Salt stress is one of the main abiotic stress factors that restrict crop productivity and quality. About 20% of irrigated farmland imposes salt stress on plant growth, with effects on crop growth [38]. Due to improper irrigation and the use of chemical fertilizer, secondary salinization of soil is becoming more problematic, reaching 2–8 g·L$^{-1}$ in some vineyards, which restricts the healthy and sustainable development of the grape industry [39]. Therefore, we determined the expression of *VvUSPs* under different concentrations of salt stress at 20 and 40 days (Figure 8). The expression level of VIT_01s0011g03680 increased significantly in response to an increase in NaCl concentration after both 20 and 40 days. Notably, the expression levels of three *VvUSPs* (VIT_07s0005g00290, VIT_10s0115g00430 and VIT_12s0134g00430) were all markedly increased at 20 days under salt treatment, indicating these *VvUSP* members are more sensitive to salt stress. With increasing salt stress time, the *VvUSPs*' sensitivity to NaCl concentration increased dynamically. In addition, the expression of two genes, VIT_01s0011g03680 and VIT_17s0000g04260, was significantly up-regulated after 40 days of salt stress, of which VIT_01s0011g03680 showed the most significant increase, suggesting this *VvUSP* may play an important regulatory role in the response to salt stress.

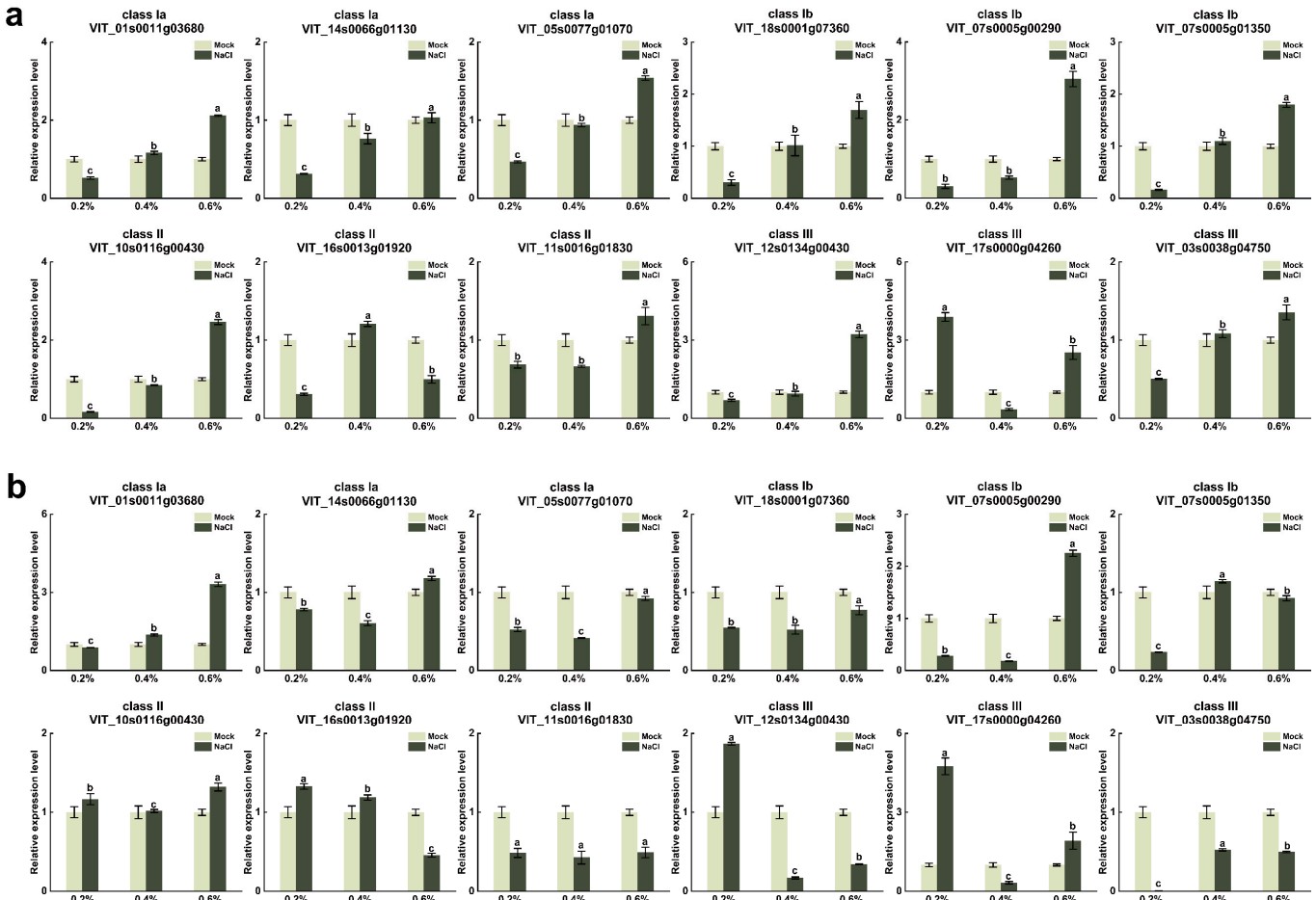

**Figure 8.** Expression levels of *VvUSP* genes at 20 days (**a**) and 40 days (**b**) of salt-stress treatments by qRT-PCR. Different letters above the bars indicate a significant difference (*p* < 0.05).

Agricultural production is severely threatened by a series of extreme climate events, including increasing temperatures [40]. Heat stress has become one of the most serious threats in viticulture [28]. Under heat stress, the expression patterns of *VvUSPs* can be categorized into four trends (Figure 9). VIT_11s0016g01830 showed a trend of significant increase over 0–6 h. In contrast, the expression levels of VIT_10s0116g00430 and VIT_16s0013g01920 showed negative correlations with increasing stress time. Eight *VvUSP* genes (83.3%) were down-regulated after 3 h of treatment. However, this decrease was followed by a significant increase in VIT_01s0011g03680, VIT_14s0056g01130, VIT_05s0077g01870, VIT_18s0001g07360, VIT_07s0005g01350, VIT_12s0134g00430 and VIT_17s0000g04260. Conversely, the expression level of VIT_03s0038g04750 increased after 3 h but decreased after 6 h. This indicates that the *VvUSP* VIT_11s0016g01830 is more sensitive to heat stress.

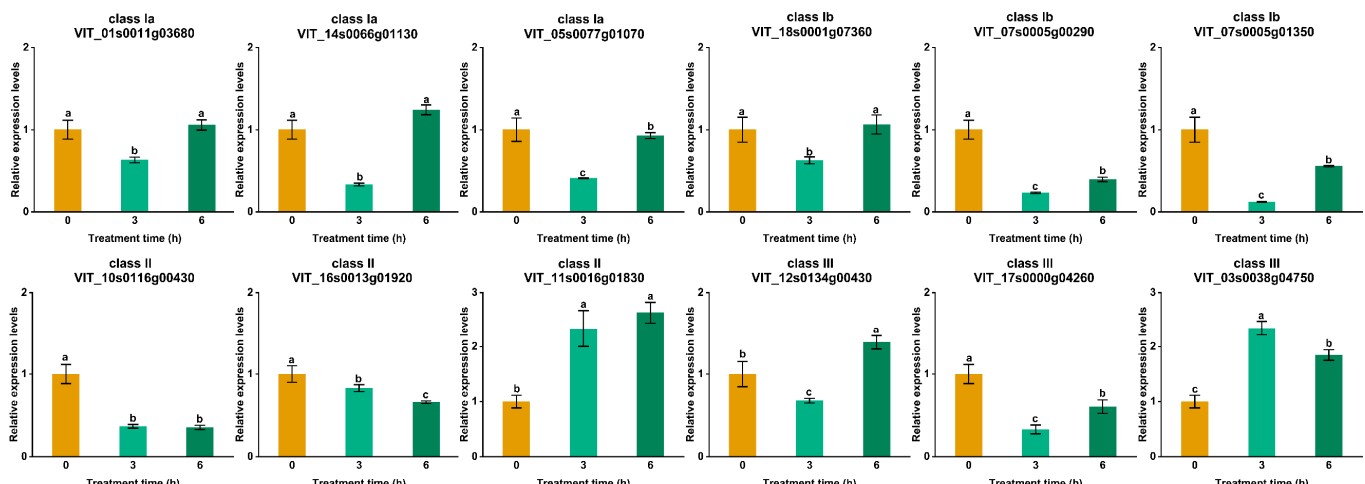

**Figure 9.** Expression levels of *VvUSP* genes under heat-stress treatments by qRT-PCR. Different letters above the bars indicate a significant difference ($p < 0.05$).

## 4. Discussion

Grapes are one of the most widely cultivated and consumed fruit species worldwide [39], but frequently suffer from various environmental stresses during cultivation, leading to substantial yield losses [29]. The *USP* gene family plays pivotal roles in the response to abiotic stresses in many plant spp., including *Arabidopsis thaliana* [29], maize [41], rice [9], *Salvia miltiorrhiza* [42] and *Gossypium* [43]. Previous studies have analyzed the changes in the expression patterns of *VvUSPA* members after drought stress and hormone treatments [13]. However, grapevine *USPs* have received little attention to date. Therefore, this study used a bioinformatic approach to identify members of the VvUSP family in the grapevine genome and to indicate potential differences in their functionalities. qRT-PCR was also used to analyze changes in *VvUSP* gene expression under abiotic stress and in response to hormonal treatments.

We identified 30 *VvUSP* genes in this work. The number of *VvUSPs* identified was similar to that in *Salvia miltiorrhiza* [42]. These VvUSPs and AtUSPs were divided into groups A, B, C and D (Figure 1). The phylogenetic analysis found that 16 VvUSPs had orthologs in Arabidopsis. Of these, VIT_04s0008g01360 and VIT_08s0062g00590 are homologous to multiple AtUSP sequences. This indicates that these VvUSPs and AtUSPs may have some similar functions. We find that the distribution of conserved sequence motifs of VvUSPs is more similar within hierarchically clustered VvUSPs than between different clusters (Figure 2a). The observed protein sequence length and isoelectric points of the VvUSPs are significantly different, which indicates that there are significant differences between *VvUSP* genes (Table 1).

Using protein sequences to predict the specific location of *VvUSP* expression products in cells is conducive to understanding the way *VvUSP* functions and understanding the role at the system level [44]. The putative localization of USP proteins was analyzed, indicating most of the proteins are localized in the cytoplasm, and suggesting that *VvUSPs* largely play important roles in cell metabolism, growth and differentiation. In addition, some were localized in the nucleus, such as VIT_17s0000g04260 and VIT_00s0203g00200, indicating they may function as transcription factors [45]. Moreover, the classification of VvUSPs was further supported by conserved motifs, exons and introns, with each subgroup sharing similar motifs, exons and introns (Figure 2b). In previous studies, genes with few or no introns were artificially expressed at relatively low levels but responded quickly to various stresses [29,46]. The expression of the intron-less gene, VIT_17s0000g04260, was seen to show rapid increases in expression in response to shade treatment (Figure 7). Furthermore, we found that 30 *VvUSP* genes were unevenly distributed on 15 chromosomes by chromosome mapping (Figure 3a). Some of the genes located on the same chromosome

were proximal, such as VIT_01s0011g03680, VIT_01s0011g03690 and VIT_01s0011g03700 on chromosome 1. From the phylogenetic tree, we observed that genes that are in proximity on the same chromosome are more likely to cluster together, whereas genes that are distally located on the same chromosome are less likely to cluster together. We predicted genes located closer on the same chromosome may have closer evolutionary relationships.

Gene family members are derived from a common ancestor, which is duplicated and diversified through chromosomal events and mutations. It is widely acknowledged that gene duplication is a driving force of gene innovation [38]. Although the main types of gene duplications can be divided into tandem and fragment replications, we found that the types of gene replication in all 6 *VvUSP* genes are fragment replication [38]. Therefore, segmental duplication appears to have been the main pathway involved in the expansion of the *VvUSP* gene family. In addition, $Ka/Ks$ value analysis in the collinearity analysis of *VvUSP* homologs shows that the values for most *VvUSP* gene pairs (5/6) are less than 1, indicating that the *VvUSP* gene family has undergone purification selection in order to eliminate harmful mutations and maintain original functions during grapevine evolution (Table 2) [38]. Interestingly, we found that AT3G62550 and AT3G53990 are not only evolutionarily related to VIT_07s0005g01350 and VIT_08s0003g01430, but are also evolutionarily related to *OsUSP* [9]. Moreover, VIT_07s0005g01350 and VIT_08s0003g01430 are also associated with VIT_05s0020g01190 and VIT_06s0004g05730, respectively. These results illustrated that the sequence variations of VIT_07s0005g01350 and VIT_08s0003g01430 increased the diversity of *VvUSP* genes.

The potential responsiveness of a gene to hormones and abiotic stresses can be assessed by the analysis of *cis*-acting elements present in its promoter. We compared *cis*-acting elements in *VvUSP* family members and found several elements, including ABREs, AREs, LTRs, MBSs and TC-rich repeats [9]. Interestingly, we found that some *VvUSPs* showed modulation under different hormone treatment conditions, of which VIT_10s0116g00430 and VIT_16s0013g01920 showed high sensitivity to MeJA treatment and contain the MeJA action element (CGGTA-motif). In addition, phytohormones play central roles in boosting plant tolerance to environmental stresses [47]. We found that the expression level of VIT_16s0013g01920 decreased significantly with the increase of NaCl concentration from 0.4% to 0.6%, whereas its expression level increased significantly after hormone treatment (Figure 8). However, plant hormones play an important role in regulating plant growth and development and resisting abiotic stress [48]. Therefore, we speculate that exogenous hormone treatment can induce *VvUSP* upregulation to resist salt stress.

Extreme weather events have become more frequent, including drought, salinity, extreme temperatures, resulting in low crop yields [49]. Shade stress restricts the rate of photosynthesis, resulting in a decrease in grape growth [50]. A number of *VvUSPs* exhibited strong up-regulation when exposed to stress from sub-optimal illumination. We observed that VIT_17s0000g04620, VIT_16s0013g01920 and VIT_11s0016g01830 were more sensitive to low light stress and were expressed at higher levels in tissues with higher photosynthetic activities, such as in leaves and flowers (Figures 5 and 7). High salinity inhibits crop growth by disrupting a range of physiological processes including photosynthesis [51]. These observations show that *VvUSPs* play an important role in protecting plants from external stress. Looking at the expression pattern of *VvUSP* genes under heat-stress conditions, VIT_11s0016g01830 play an important role in improving the tolerance of plants to heat stress.

## 5. Conclusions

In this study, a genome-wide analysis of the grape *USP* gene family was performed, and 30 *VvUSPs* were identified. The chromosomal localization, phylogenetic tree, collinearity, gene structure, conserved motif and *cis*-acting elements of these genes were analyzed by bioinformatic approaches to reveal the evolutionary mechanisms underlying the *VvUSP* gene family in grape. In addition, the analysis of the expression patterns of *VvUSPs* demonstrated their differential tissue-specificity and sensitivities to SA and MeJA hor-

mone treatments, and to abiotic stresses. In summary, this study provides comprehensive information that will be useful in the further investigation into the genetics and protein functions of the *USP* gene family in grapevine.

**Supplementary Materials:** The following are available online at https://www.mdpi.com/article/10 .3390/horticulturae8111024/s1, Table S1: The primer sequences for qRT-PCR; Table S2. List of *USP* coding sequences in grape; Figure S1: Conserved motifs analysis in *VvUSP* gene family; Table S3. The sequences of ten conserved motifs; Table S4: *Cis*-acting components of *USP* genes in grape.

**Author Contributions:** T.X. wrote the manuscript; Y.W. and J.Y. were involved in the experimental design. T.X., T.C., T.Z., L.S., Z.C. and Y.X. were involved in the collection and analysis of data. All authors have read and agreed to the published version of the manuscript.

**Funding:** This research was funded by the Key Research and Development Program of Zhejiang Province (2021C02053) and 2025 Major Science and Technology Innovation Special Project of Ningbo (2019B10015).

**Institutional Review Board Statement:** Not applicable.

**Informed Consent Statement:** Not applicable.

**Data Availability Statement:** Not applicable.

**Conflicts of Interest:** The authors declare that they have no conflict of interest.

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
