# Peer review of "Genome-Wide Identification and Characterization of USP Gene Family in Grapes (Vitis vinifera L.)"

_horticulturae, doi:10.3390/horticulturae8111024_

Round 1
Reviewer 1 Report
Introduction:
-Line 59: Include gibberellin (GA). “A central role for the gibberellin (GA) class of growth hormones in the response to abiotic stress is becoming increasingly evident. Reduction of GA levels and signalling has been shown to contribute to plant growth restriction on exposure to several stresses, including cold, salt and osmotic stress.” Colebrook EH, Thomas SG, Phillips AL, Hedden P. The role of gibberellin signalling in plant responses to abiotic stress. J Exp Biol. 2014 Jan 1;217(Pt 1):67-75. doi: 10.1242/jeb.089938. PMID: 24353205.
Materials and methods:
- Line 82: please indicate the genome version of grape and Arabidopsis thaliana, is important for the reproducibility of the experiment.
-Line 87: the Pfam website is now host by InterPro, you should change the URL ( InterPro is the new home of Pfam The Pfam website (pfam.xfam.org) was shut down on October 5th, but InterPro offers the same functionality and data. A legacy version of Pfam will remain available at https://pfam-legacy.xfam.org/ until January 2023, but will not receive any updates. You can read more about our decision to shut down the Pfam website in our blog post. )
Line 124: Please indicate the product name or were is purchased.
Line 126: What are the light/temperature conditions in the greenhouse? Explain better the parameter that indicate that you have 0%, 30%, 60% and 90% transmittance.
Author Response
Response to Reviewer 1 Comments
Point 1:
Line 59: Include gibberellin (GA). “A central role for the gibberellin (GA) class of growth hormones in the response to abiotic stress is becoming increasingly evident. Reduction of GA levels and signalling has been shown to contribute to plant growth restriction on exposure to several stresses, including cold, salt and osmotic stress.” Colebrook EH, Thomas SG, Phillips AL, Hedden P. The role of gibberellin signalling in plant responses to abiotic stress. J Exp Biol. 2014 Jan 1;217(Pt 1):67-75. doi: 10.1242/jeb.089938. PMID: 24353205.
Response 1:
"gibberellin (GA)" have been added
Point 2:
Line 82: please indicate the genome version of grape and Arabidopsis thaliana, is important for the reproducibility of the experiment.
Response 2:
The genome version of grape and Arabidopsis thaliana has been indicated
Point 3:
Line 87: the Pfam website is now host by InterPro, you should change the URL (InterPro is the new home of Pfam The Pfam website (pfam.xfam.org) was shut down on October 5th, but InterPro offers the same functionality and data. A legacy version of Pfam will remain available at https://pfam-legacy.xfam.org/ until January 2023, but will not receive any updates. You can read more about our decision to shut down the Pfam website in our blog post.)
Response 3:
"http://pfam.janelia.org/ " has been replace to "http://pfam-legacy.xfam.org/"
Point 4:
Line 124: Please indicate the product name or were is purchased.
Response 4:
"SA" and "MeJA" product name have been added.
Point 5:
Line 126: What are the light/temperature conditions in the greenhouse? Explain better the parameter that indicate that you have 0%, 30%, 60% and 90% transmittance.
Response 5:
The light/temperature conditions in the greenhouse have been added. 0%, 30%, 60% and 90% transmittance has been indicated.

Reviewer 2 Report
The manuscript “Genome-Wide Identification and Characterization of USP Gene Family in Grapes (Vitis vinifera L.)” by Xu et al., have made a good attempt at genome-wide study and prediction of USP gene family in grapes, an important economical crop. Given that the involvement of USP gene family in growth and development, abiotic and biotic stress, this study provides a compelling prediction of grape USP genes and their probable function in shade, salt, and heat stress. The findings of this research will address many new researchers in this field. The authors have performed the experiment in a systematic manner giving all the details required for the experiment. The authors have discussed the results with recent researchers related to this theme. The conclusion is crisp and concise. The language used in the manuscript is standard scientific English language which is easier for everyone to understand however some grammatical errors could be rectified. Considering all the above points, I recommend acceptance of the manuscript with few suggestions.
1. Line 148- remove coding sequence size.
2. Line 150. Full stop.
3. Follow similar pattern for writing gene and protein names.
4. Explain the figure legends properly.
5. Mention internal control and relative expression in qRT-PCR material and method section.
Author Response
Response to Reviewer 2 Comments
Point 1:
Line 148- remove coding sequence size.
Response 1:
"Coding sequence size" has been deleted.
Point 2:
Line 150. Full stop.
Response 2:
"." have been added in the line 150.
Point 3:
Follow similar pattern for writing gene and protein names.
Response 3:
Gene and protein names have been standardized.
Point 4:
Explain the figure legends properly.
Response 4:
Figure legends has been modified and improved.
Point 5:
Mention internal control and relative expression in qRT-PCR material and method section.
Response 5:
qRT-PCR reaction system and reaction program have been standardized.
